# A Prognostic Merit of Statins in Patients with Chronic Hemodialysis after Percutaneous Coronary Intervention—A 10-Year Follow-Up Study

**DOI:** 10.3390/jcm11020390

**Published:** 2022-01-13

**Authors:** Takehiro Funamizu, Hiroshi Iwata, Yuichi Chikata, Shinichiro Doi, Hirohisa Endo, Hideki Wada, Ryo Naito, Manabu Ogita, Yoshiteru Kato, Iwao Okai, Tomotaka Dohi, Takatoshi Kasai, Kikuo Isoda, Shinya Okazaki, Katsumi Miyauchi, Tohru Minamino

**Affiliations:** 1Department of Cardiovascular Biology and Medicine, Juntendo University Graduate School of Medicine, Tokyo 113-8421, Japan; t-funamizu@juntendo.ac.jp (T.F.); ychikata@juntendo.ac.jp (Y.C.); doies@juntendo.ac.jp (S.D.); hendo@juntendo.ac.jp (H.E.); rnaitou@juntendo.ac.jp (R.N.); y-0104@juntendo.ac.jp (Y.K.); okaiwao@juntendo.ac.jp (I.O.); tdohi@juntendo.ac.jp (T.D.); tkasai@juntendo.ac.jp (T.K.); kisoda@juntendo.ac.jp (K.I.); shinya@juntendo.ac.jp (S.O.); ktmmy@juntendo.ac.jp (K.M.); t.minamino@juntendo.ac.jp (T.M.); 2Department of Cardiology, Juntendo University Shizuoka Hospital, Izunokuni 410-2295, Japan; hdwada@juntendo.ac.jp (H.W.); m-ogita@juntendo.ac.jp (M.O.)

**Keywords:** hemodialysis, statin, secondary prevention, percutaneous coronary intervention, cardiovascular death

## Abstract

Background: Patients with end-stage renal disease (ESRD) on chronic hemodialysis who are complicated by coronary artery disease (CAD) are at very high risk of cardiovascular (CV) events and mortality. However, the prognostic benefit of statins, which is firmly established in the general population, is still under debate in this particular population. Methods: As a part of a prospective single-center percutaneous coronary intervention (PCI) registry database, this study included consecutive patients on chronic hemodialysis who underwent PCI for the first time between 2000 and 2016 (*n* = 201). Participants were divided into 2 groups by following 2 factors, such as (1) with or without statin, and (2) with or without high LDL-C (> and ≤LDL-C = 93 mg/dL, median) at the time of PCI. The primary endpoint was defined as CV death, and the secondary endpoints included all-cause and non-CV death, and 3 point major cardiovascular adverse events (3P-MACE) which is the composite of CV death, non-fatal myocardial infarction and stroke. The median and range of the follow-up period were 2.8, 0–15.2 years, respectively. Results: Kaplan–Meier analyses showed significantly lower cumulative incidences of primary and secondary endpoints other than non-CV deaths in patients receiving statins. Conversely, no difference was observed when patients were divided by the median LDL-C at the time of PCI (*p* = 0.11). Multivariate Cox proportional hazard analysis identified statins as an independent predictor of reduced risk of CV death (Hazard ratio of statin use: 0.43, 95% confidence interval 0.18–0.88, *p* = 0.02), all-cause death (HR: 0.50, 95%CI 0.29–0.84, *p* = 0.007) and 3P-MACE (HR: 0.50, 95%CI 0.25–0.93, *p* = 0.03). Conclusions: Statins were associated with reduced risk of adverse outcomes in patients with ESRD following PCI.

## 1. Introduction

In accordance with the increase of the global prevalence of chronic kidney disease (CKD) as a major healthcare burden worldwide [1], the number of patients with end stage renal disease (ESRD) requiring chronic renal replacement therapy, including hemodialysis, peritoneal dialysis and functioning kidney transplantation, has significantly increased over time [2,3]. Among patients with CKD, which is associated with substantially higher cardiovascular (CV) mortality and morbidity rates [4], those on chronic hemodialysis are at particularly very high risk of all-cause as well as CV mortality [5]. In patients on chronic hemodialysis, coronary artery disease (CAD) is a major concern due to its substantially high prevalence at the induction of maintenance hemodialysis therapy [6]. Patients on hemodialysis and with established CAD following percutaneous coronary intervention (PCI) are much more prone to subsequent repeated revascularization and CV death [7]. Therefore, optimal medications for secondary prevention of CV events are critically important in patients with both chronic hemodialysis and CAD. The accumulating evidence has obviously indicated the effectiveness of statins in the primary and secondary prevention of CV diseases in diverse high-risk populations, such as CKD patients [8]. However, three previous landmark randomized trials involving patients with maintenance hemodialysis [9,10,11] have failed to show the benefit of statins at reducing the risk of cumulative CV events and all-cause mortality, and current guidelines have not recommended the initiation of statins for hemodialysis patients [12]. Nevertheless, since these studies did not include many patients having both ESRD on dialysis and CAD, the merit of statins is still open to question in such particular population.

Therefore, the aim of this observational study involving patients in a prospective PCI registry database was to explore in detail whether statin therapy was associated with a reduced risk of adverse CV outcomes in hemodialysis patients following PCI.

## 2. Materials and Methods

This study was performed in accordance with the Declaration of Helsinki and with approval from the Institutional Review Board (IRB) of Juntendo University (IRB number: 17-170). The prospective registry database of patients who underwent any PCI at Juntendo University Hospital, Tokyo, Japan (Juntendo Physicians’ Alliance for Clinical Trial: J-PACT) is publicly registered (University Medical Information Network Japan-Clinical Trials Registry, UMIN-CTR 000035587). Written informed consent was obtained from all participants for the J-PACT registry.

### 2.1. Participants and Follow-Up Duration

#### 2.1.1. Participants

This study is a retrospective analysis of a portion of a prospective single-center registry database of patients who underwent PCI at Juntendo University Hospital. The database was launched in February 1984 (Juntendo Physicians’ Alliance for Clinical Trial, J-PACT). The registry database includes data regarding patient demographics, coronary artery lesions, PCI procedures, and devices used during the procedure. Patients who underwent any type of percutaneous coronary artery intervention procedure, including thrombectomy, balloon angioplasty, and/or deployment of any type of coronary stent, were enrolled in the registry. In the study period between 2000 and 2016, consecutive 4542 patients who underwent PCI for the first time were registered in the database. After excluding patients without maintenance hemodialysis at PCI (*n* = 4341), the remaining 201 patients were enrolled in the present study. Patients with ESRD who underwent temporary hemodiafiltration and peritoneal dialysis were excluded. Thereafter, we divided the participants into 2 groups according to whether they did or did not receive statins at PCI (Statin group and Non-statin group, *n* = 73 and *n* = 128, respectively) (Appendix A). Blood samples were collected in the early morning after overnight fasting.

#### 2.1.2. Follow-Up

In this prospective PCI registry database, the follow-up of patients was based on chart review, as far as they were followed at Juntendo University Hospital. A prognosis survey questionnaire was mailed out every 5 years if they were transferred to other institutions. When there was no response to the questionnaire, further follow-up information was obtained by telephone. In cases in which no response could be obtained by either mail and phone, follow-up was terminated at the latest time point at which their survival at our institution was confirmed, such as the last visit date to an outpatient clinic or the last day of any hospitalization. The median and range of the follow-up period since the index PCI were 2.8 and 0–15.2 years, respectively.

#### 2.1.3. Endpoints

The primary endpoint set in the present study was CV death, which was defined as a composite of the following types of death; sudden death in which non-cardiac death could not be excluded, and death due to myocardial infarction, heart failure, cardiogenic shock, a cerebrovascular event, or aortic diseases. The secondary endpoints included all-cause death, mortality other than cardiovascular-related (non CV death) and 3 point major adverse cardiovascular events (3P-MACE) defined as a composite of CV death, non-fatal myocardial infarction, and non-fatal stroke [13].

### 2.2. Statistical Analysis

Continuous variables are presented as the mean ± standard deviation or median with interquartile range (IQR) in accordance with the results of the Shapiro–Wilk normality test. Categorical variables are presented as the actual number and frequencies (%). Quantitative data across groups were compared using the ANOVA test or the Kruskal–Wallis test. Categorical variables were compared using the Fisher-exact test with the chi-squared test. Unadjusted Kaplan–Meier analysis evaluated the time to the cumulative CV mortality, all-cause mortality, 3P-MACE, and non-CV mortality followed by the log-rank test for comparisons. The prognostic impact of statins at PCI on CV death, all-cause death and 3P-MACE were assessed by univariate and multivariate Cox proportional hazards regression analyses calculating hazard ratios (HR) with 95% confidence intervals (CI) for three endpoints. For multivariate analyses, covariates other than statin therapy at PCI were selected based on the findings of baseline characteristics and univariate unadjusted analyses in addition to previously established factors associated with vital consequences in this population, such as age, gender and diabetic nephropathy [14,15]. Covariates used in 3 models are summarized in Appendix A. Model 1 included age, gender, BMI and diabetic nephropathy in addition to statin therapy. Model 2 had beta-blockers and LDL-C level in addition to the covariates in Model 1, and Model 3 had hs-CRP in addition to the covariates of Model 2. Age, BMI, LDL-C and hs-CRP were assessed as continuous variables in all models. In multivariate Cox proportional hazard models, interactions amongst statin therapy and LDL-C levels were assessed (Appendix A). A *p*-value (*p*) < 0.05 was considered to indicate statistical significance. Statistical analyses were performed using JMP version 12.1 (SAS Institute, Cary, NC, USA) and SPSS version 26 (IBM Corp., Armonk, NY, USA).

## 3. Results

### 3.1. Baseline Demographics and Procedural Characteristics

The present study included 201 patients with chronic hemodialysis who underwent PCI. The mean age was 66.0 ± 10.3 years, and the median duration of hemodialysis was 5.2 years. Statins were prescribed in 73 patients (36.3%) (Statin group) and not prescribed in 128 patients (Non-statin group). In the statin group, 51 (69.9%) received low-intensity and 22 (30.1%) moderate-intensity statins, while none received high-intensity statins according to the guidelines, atorvastatin 40–80 mg/day and rosuvastatin 20–40 mg/day [16,17]. The baseline and procedural characteristics are listed and compared in Table 1.

Both groups were similar in most of the demographic and procedural factors, except for a significantly lower proportion of males and lower LDL-C levels in the Statin group.

There were no significant differences between the groups regarding medications, including beta-blockers, calcium channel blockers (CCBs), angiotensin converting enzyme (ACE) inhibitors, and angiotensin receptor blockers (ARB).

### 3.2. Reduced Rates of Adverse Events Following PCI in Patients Receiving Statins, While There Was No Difference in Groups Divided by Preprocedural LDL-C Level

During the entire follow-up period, all-cause deaths occurred in 86 out of 201 patients (42.8%), including 47 CV deaths (23.4%) and 39 non-CV deaths (19.4%), and 3P-MACE were identified in 58 patients (28.9%). Compared to the Non-statin group, the incidences per 1000 person-years of CV death, all-cause death and 3P-MACE were significantly lower in the Statin group, whereas those of non-CV death were not different in two groups (Table 2).

Unadjusted Kaplan–Meier analyses followed by the log-rank comparison test consistently showed significantly lower cumulative cardiovascular-related and all-cause mortality rates, as well as a lower incidence of 3P-MACE in the Statin group (*p* = 0.007, 0.004 and 0.016 by log-rank comparison, respectively) (Figure 1A–C), while similar cumulative non-CV mortalities in both groups (*p* = 0.18) (Figure 1D).

In contrast, unadjusted Kaplan–Meier analysis revealed no significant difference in any cumulative incidences of CV (Figure 2A), all-cause (Figure 2B), 3P-MACE (Figure 2C) and non-CV mortalities (Figure 2D), when participants were divided by the median of LDL-C (93 mg/dL).

### 3.3. Adjusted Prognostic Impact of Statin Use for Outcomes Following PCI in Patients with Maintenance Hemodialysis

To further address the prognostic effect of statin therapy in patients with chronic hemodialysis after PCI, univariate-unadjusted and multivariate-adjusted Cox proportional hazard analyses were used to calculate the hazard ratios (HRs) of statin therapy for CV death, all-cause death and 3P-MACE, respectively. Covariates included in the multivariate models are summarized in Appendix A. Most of the interactions among statin therapy and LDL-C in Models 2 and 3 for CV death, all-cause death and 3P-MACE were not significant (*p* > 0.05), except statin therapy and LDL-C in Model 3 for all-cause death (*p* = 0.01) (Appendix A). Univariate analysis showed that statin therapy was significantly associated with reduced risk of all endpoints (Appendix A). Multivariate analyses using a model including conventional risk factors in the population after PCI (Model 1) showed a substantial risk reduction by statin therapy for all endpoints (HR and 95% confidence interval (CI) (*p*-value) for CV death: 0.37, 0.16–0.75 (0.005), for all-cause death: 0.47, 0.27–0.78 (0.003) and for 3P-MACE: 0.46, 0.23–0.85 (0.01), respectively). Furthermore, analyses using a different model which additionally included receiving beta blockers at PCI and serum level of LDL-C (Model 2) calculated significantly reduced HRs for all endpoints comparable with Model 1. In a model that further included serum hs-CRP level in addition to the variates of Model 2 (Model 3), a risk reduction by statin therapy for CV death and all-cause death was observed, and it also showed the similar tendency for 3P-MACE, although the difference was not statistically significant (0.53, 0.26–1.00 (0.05), respectively) (Figure 3).

## 4. Discussion

The present observational study involving consecutive 201 patients with ESRD requiring hemodialysis who underwent PCI demonstrated that statin therapy at PCI was significantly associated with reduced risk of subsequent CV and all-cause mortalities and 3P-MACE, while it had no effect on non-CV mortalities. In contrast, the cumulative incidences of CV death, all-cause death and 3P-MACE were similar in the two groups divided by the median LDL-C (93 mg/dL). Furthermore, statins were associated with substantially reduced risk of CV death, all-cause death and 3P-MACE after adjusting for gender, age, diabetic nephropathy, BMI, receiving beta blockers, hs-CRP, and LDL-C level.

The number of patients with ESRD who are on chronic hemodialysis is increasing worldwide [2,3,18]. Although the age distribution trend indicates an increase in older patients among those with chronic hemodialysis, overall mortality has declined in the past 20 to 30 years [2,19,20]. The appropriate control of risk factors in patients with dialysis, such as blood pressure [21] and blood glucose [22], as well as the technical advancements in dialysis may have contributed to lowering the mortality in this population. Numerous lines of evidence have established the efficacy of statin therapy in the primary and secondary prevention of atherosclerotic CV diseases for the entire, and wide range of subpopulations at high risk, including those with CKD [8,23]. However, previous randomized trials and a meta-analysis of a dialysis population have shown little or no effect of statins for improving outcomes, and even if they have any favorable prognostic effect, the magnitude of the relative reduction of statins in the ESRD population appears to be substantially smaller than in those who are in earlier stages of CKD [9,10,11,24]. Accordingly, current guidelines of lipid management have recommended that statin therapy should not to be initiated for patients with ESRD requiring dialysis [12,16,17,25]. Moreover, there is a paucity of data regarding whether statins have merit in the population of patients with both ESRD on dialysis and with established CAD and who underwent coronary revascularization. Many of the randomized CV outcome trials of statins for the secondary prevention of coronary artery disease have excluded individuals with dialysis [26,27], and the number of patients who had a history of coronary revascularization in previous statin trials involving dialysis patients was limited (13% in 4D and 6.2% in AURORA) [9,10]. Additionally, there is an emerging concern that statins might enhance arterial calcification in patients with CKD [28,29]. Conversely, several observational studies utilizing a large-scale registry database have reported the continuation of statin therapy through the induction of maintenance dialysis in patients with ESRD was associated with lower all-cause and CV mortality rates, compared to individuals in whom it was discontinued through the pre- to post-dialysis stages [30,31]. Accordingly, despite no recommendation concerning the initiation of statins, guidelines paradoxically encourage the continuation of statins in patients who were already receiving them before the introduction of dialysis [12,16,17,25]. Regarding patients with both ESRD and CAD, two observational studies from east Asia have indicated a prognostic merit of statins is to reduce the risk of all-cause mortality following myocardial infarction [32] and the composite of non-fatal myocardial infarction, stroke and all-cause mortality following PCI [33]. However, the overall quality of evidence to guide a therapeutic strategy in this specific population is still insufficient, although the rates of mortality and adverse CV events following PCI are extremely high [34,35,36]. The present study showed the significant prognostic merit of statin therapy since it was associated with reduced risk of CV and all-cause mortality and 3P-MACE following PCI. Moreover, we evaluated the merit of statins in this population in light of LDL-C, a main target of statins, which has rarely been assessed thus far [32,33]. In the present study, the Kaplan–Meier curves of groups with and without high LDL-C (> and ≤median, 93 mg/dL) were not significantly different. In multivariate Cox proportional hazard analyses, the significant risk reduction by statins was maintained, when it was adjusted by LDL-C level for all endpoints. These findings suggest that the prognostic merit of statins was not mainly mediated by LDL-C lowering property of statins. Moreover, the prognostic impact of statins was further adjusted by using multivariate models including previously established risk factors in the population of the present study, such as reduced BMI and elevated hs-CRP, corresponding to poor nutritional status [37,38] and continuous systemic inflammation [39], respectively [40,41,42]. Even after adjustment for these risk factors, the prognostic merit of statins in this study for all three endpoints remained significant. Taken together, our findings indicated that statins might be effective for better long-term outcomes in ESRD patients complicated by CAD, and the benefit was not mainly mediated by its LDL-C lowering property. Accordingly, the overall findings of this study may suggest the need to encourage not only continuing, but also initiating, statin therapy in dialysis patients complicated by CAD. The significance of this study which might add novel evidence, not negate previous findings, is that there is a prognostic merit of statins in the specific population of maintenance HD complicated by established CAD and the benefit was through the mechanism other than LDL-C lowering property of statins.

This study has several limitations. First, since it was retrospective in nature due to its analysis of a single center prospective registry database involving a relatively small number of participants without any randomization, and unaccounted confounding factors, which were not recorded or were not even included in the model, may potentially induce bias, although we have adjusted for known confounding factors. Second, the duration and dose of the statin pretreatment before the PCI procedure were not determined, and they may have had dominant effects on the outcomes in this population. Third, the adverse effects of statin treatments were not evaluated in detail. Despite these limitations, the evaluation of the cause of death, patient background data including blood sampling data, such as LDL-C and hs-CRP, and the long-term follow-up period have strengthened the significance of the findings of the present study.

## 5. Conclusions

The present study has shown that statin administration was significantly associated with better outcomes in patients with ESRD on chronic hemodialysis following PCI. Although the mechanisms of the prognostic merit of statins are yet to be elucidated, the study results indicate that it might be independent from LDL-C lowering property of statins.

## Figures and Tables

**Figure 1 jcm-11-00390-f001:**
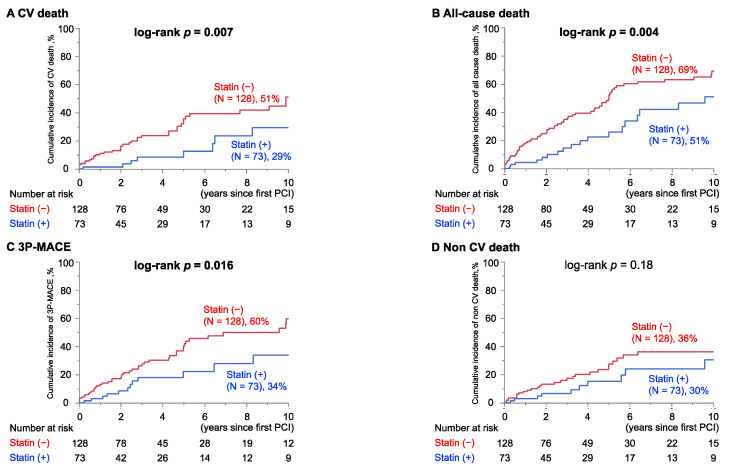
Cumulative cardiovascular event rates in groups with and without statins. Significantly reduced cumulative incidences of (**A**) CV death, (**B**) All-cause death, and (**C**) 3P-MACE were observed in patients with statin therapy at PCI, while no difference in (**D**) Non CV death was seen. Percent indicates the cumulative incidence of events at 10 years of follow-up in each group.

**Figure 2 jcm-11-00390-f002:**
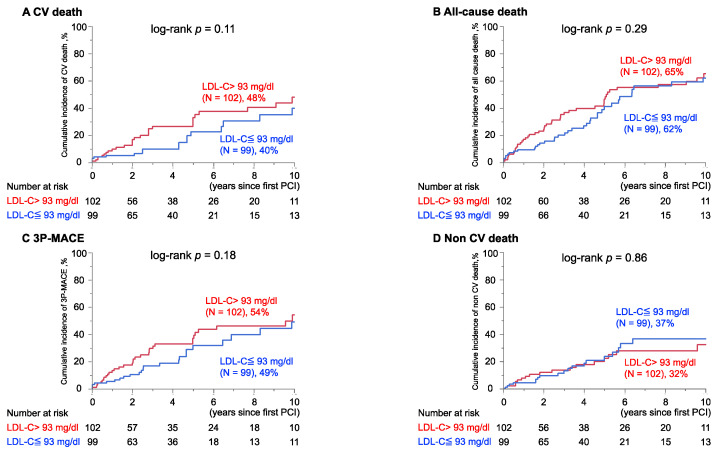
Cumulative cardiovascular event rates in groups divided by median LDL-C at procedure. Similar cumulative incidences of (**A**) CV death, (**B**) All-cause death, (**C**) 3P-MACE and (**D**) Non CV death in patients with and without LDL-C higher than median (>93 mg/dL). Percent indicates the cumulative incidence of events at 10 years of follow-up in each group.

**Figure 3 jcm-11-00390-f003:**
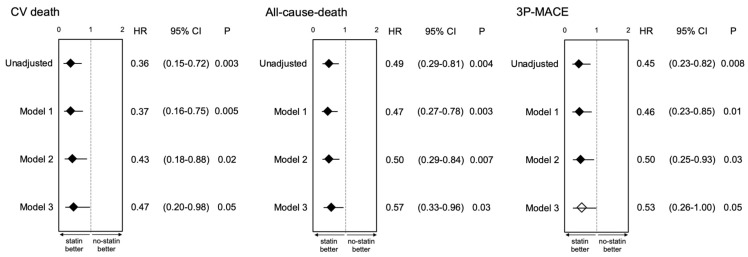
Hazard ratios of statins for cardiovascular events in hemodialysis patients following PCI. Hazard ratios of statin therapy at PCI for unadjusted and adjusted by 3 models. Closed and open rhombuses indicate significant (*p* < 0.05) and insignificant (≥0.05) association of statin therapy with endpoints, respectively.

**Table 1 jcm-11-00390-t001:** Baseline clinical characteristics of the study population.

	Overall (*n* = 201)	Statin (*n* = 73)	Non-Statin (*n* = 128)	*p*-Value
BMI ^1^, kg/m^2^	22.4 ± 3.9	22.6 ± 4.3	22.3 ± 3.6	0.67
Age, years	66.0 ± 10.3	65.2 ± 11.9	66.6 ± 9.3	0.35
Male, *n* (%)	168 (83.6)	56 (76.7)	112 (87.5)	0.047
Hypertension, *n* (%)	174 (86.6)	60 (82.2)	114 (89.1)	0.17
Dyslipidemia, *n* (%)	156 (77.6)	73 (100)	83 (64.8)	<0.001
Diabetic nephropathy, *n* (%)	123 (61.2)	44 (60.3)	79 (61.7)	0.84
Duration of hemodialysis (years)	5.2 (1.6–10.0)	4.0 (1.2–8.9)	5.4 (1.9–10.9)	0.20
History of smoking, *n* (%)	118 (58.7)	39 (53.4)	79 (61.7)	0.25
Family history, *n* (%)	46 (22.9)	15 (20.6)	31 (24.2)	0.55
Atrial fibrillation, *n* (%)	26 (12.9)	8 (11.0)	18 (14.1)	0.53
Prior PCI ^2^, *n* (%)	39 (19.5)	14 (19.2)	25 (19.7)	0.93
Prior myocardial infarction, *n* (%)	44 (21.9)	17 (23.3)	27 (21.1)	0.72
Prior CABG ^3^, *n* (%)	38 (18.9)	12 (16.4)	26 (20.3)	0.50
Peripheral arterial disease, *n* (%)	55 (27.4)	21 (28.8)	34 (26.6)	0.74
Prior cerebrovascular disease, *n* (%)	34 (16.9)	11 (15.1)	23 (18.0)	0.60
ACS ^4^, *n* (%)	36 (17.9)	12 (16.4)	24 (18.8)	0.68
Number of vessels	2.1 ± 0.8	2.0 ± 0.8	2.1 ± 0.8	0.49
RCA ^5^, *n* (%)	74 (36.8)	27 (37.0)	47 (36.7)	0.97
LAD ^6^, *n* (%)	80 (39.8)	28 (38.4)	52 (40.6)	0.75
LCX ^7^, *n* (%)	38 (18.9)	15 (20.6)	23 (18.0)	0.65
LMT ^8^, *n* (%)	7 (3.5)	2 (2.7)	5 (3.9)	1.00
SVG ^9^, *n* (%)	2 (1.0)	1 (1.4)	1 (0.8)	1.00
Stent diameter, mm	3.0 ± 0.4	2.9 ± 0.4	3.0 ± 0.4	0.15
Total stent length, mm	23 (15–32)	24 (15–33)	20 (15–32)	0.25
Medications				
Beta blocker, *n* (%)	96 (47.8)	39 (53.4)	57 (44.5)	0.22
CCB ^10^, *n* (%)	105 (52.2)	34 (46.6)	71 (55.5)	0.22
ACEI/ARB ^11^, *n* (%)	106 (52.7)	33 (45.2)	73 (57.0)	0.11
Statin type				<0.001
Low-intensity statin, *n* (%)	51 (25.4)	51 (69.9)	0 (0)	
Moderate-intensity statin, *n* (%)	22 (11.0)	22 (30.1)	0 (0)	
High-intensity statin, *n* (%)	0 (0)	0 (0)	0 (0)	
Labolatory findings				
TC ^12^, mg/dL	163.3 ± 35.5	158.9 ± 34.7	165.8 ± 35.8	0.18
LDL-C ^13^, mg/dL	96.9 ± 30.3	89.6 ± 27.3	101.0 ± 31.3	0.01
HDL-C ^14^, mg/dL	42.6 ± 14.0	44.3 ± 13.4	41.6 ± 14.2	0.18
TG ^15^, mg/dL	106.0 (79.0–146.0)	107.0 (82.5–154.0)	105.5 (78.0–142.5)	0.68
Non-FBG ^16^, mg/dL	110.3 ± 41.1	104.5 ± 27.5	113.6 ± 46.8	0.13
Hemoglobin, g/dL	10.6 ± 1.5	10.9 ± 1.5	10.5 ± 1.4	0.07
HbA1c ^17^, %	6.1 ± 1.1	6.1 ± 1.0	6.1 ±1.2	0.87
Ca, mg/dL	9.0 ± 0.9	9.0 ± 1.0	9.1 ± 0.8	0.83
P, mg/dL	5.2 ± 1.4	5.0 ± 1.4	5.3 ± 1.4	0.28
hs-CRP ^18^, mg/L	3.0 (1.0–12.1)	2.1 (0.9–10.5)	3.6 (1.2–12.3)	0.13
Albumin, mg/dL	3.5 ± 0.5	3.6 ± 0.4	3.5 ± 0.5	0.18
eGFR ^19^, mL/min/1.73 m^2^	6.0 ± 2.1	6.0 ± 2.1	6.0 ± 2.2	0.97

Abbreviations: ^1^: BMI, body mass index; ^2^: PCI, percutaneous coronary intervention; ^3^: CABG, coronary artery bypass grafting; ^4^: ACS, acute coronary syndrome; ^5^: RCA, right coronary artery; ^6^: LAD, left anterior descending coronary artery; ^7^: LCX, left circumflex coronary artery; ^8^: LMT, left main trunk coronary artery; ^9^: SVG, saphenous vein graft; ^10^: CCB, calcium channel blocker; ^11^: ACEI/ARB, angiotensin-converting enzyme inhibitor/angiotensin receptor blocker; ^12^: TC, total cholesterol; ^13^: LDL-C, low density lipoprotein-cholesterol; ^14^: HDL-C, high density lipoprotein-cholesterol; ^15^: TG, triglycerides; ^16^: Non-FBG, non-fasting blood glucose; ^17^: HbA1c, glycated hemoglobin; ^18^: hs-CRP, high-sensitivity C-reactive protein; ^19^: eGFR, estimated glomerular filtration rate.

**Table 2 jcm-11-00390-t002:** Overall incidence of cardiovascular events (per 1000 person-years).

	Overall (*n* = 201)	Statin (*n* = 73)	Non-Statin (*n* = 128)	*p*-Value
All-cause death, *n* (/1000 person-years)	86 (107)	19 (64.7)	67 (131)	<0.001
Cardiovascular death, *n* (/1000 person-years)	47 (58.3)	8 (27.2)	39 (76.2)	0.002
Non-cardiovascular death, *n* (/1000 person-years)	39 (48.4)	11 (37.4)	28 (54.7)	0.24
3P-MACE ^1^, *n* (/1000 person-years)	58 (77.0)	12 (43.7)	46 (96.1)	0.003

Abbreviations: ^1^: 3P-MACE, 3 point major adverse cardiovascular events.

## Data Availability

Data available on request due to restrictions, e.g., privacy or ethical.

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
