# Peer review of "A Prognostic Merit of Statins in Patients with Chronic Hemodialysis after Percutaneous Coronary Intervention—A 10-Year Follow-Up Study"

_jcm, 2022, doi:10.3390/jcm11020390_

Round 1

Reviewer 1 Report

Funamizu et al. present an interesting single-centre retrospective analysis of 201 patients with ESRD on haemodialysis and their associated outcomes with and without statin therapy following PCI. The patient information was obtained from a PCI registry database and includes a median 2.8 year follow-up. The manuscript is well prepared and addresses an important clinical question.

Abstract

Line 17 – underwent PCI

Line 18 - I understand what you did from reading the methodology but you say 2 groups and then list 4

Line 19 – at the time of PCI

Line 20 – not that clear

Line 25 – grammar

Line 38 – which is associated with

Line 54 – check grammar

Supplementary Figure S1 – I would be interested to see the loss to follow up (your median follow up was 2.8 year to and range 0-15.2 can be broken down) numbers on the CONSORT diagram. For consistency, first box of CONSORT, should be (n=4542)

Methods Page 2, Line 83 - Follow up – was compliance to statin therapy assessed? I would think that this is of high importance provide the paper is investigating outcomes with and without statin therapy. Furthermore, were the duration of anti-platelet therapy/warfarin following PCI from the two groups collected? 

Results Page 3, Line 132 – were data collected on the distributions on the types of statins used in the statin group? there has been landmark trials for specific statins so does your population differ from those previously published

Discussion Line 271 - I think it is important to be explicit on how this study is different to previous landmark papers. Otherwise without that context, it's hard to understand why an observational study would alter guidelines that have been informed by randomised trials. My interpretation of data presented is that patients who have ESRD on HD that have PCI might benefit from statin (and this observational study should promote a RCT to confirm this retrospective finding). Also, more broadly speaking, patients where statin therapy is indicated as part of medical management for diagnosed CAD/dyslipidaemia, should possibly still be indicated for that therapy and commenced on this despite being on HD. I don't think this study should negate the findings of the other trials that would suggest that statin therapy shouldn't be initiated for the indication of ESRD(and its associated CV risk) in itself.

My understanding is that the Wanner, atorvastatin trial was also informed by positive results from observational study of the US renal data system, yet the RCT was negative.

Also the SHARP trial specifically excluded patient with CAD.

Therefore, its important to clearly explain where this study fits within current evidence and its limitations

Discussion Page 10, Line 275 – I would also think that the fact that this study was a retrospective analysis of database information from a single-centre study may introduce bias. I would think that a study with this methodology might include more discussion of limitations. 

Author Response

The authors sincerely thank the Editors and Reviewers for providing us the opportunity to revise our study, as well as for a number of insightful, helpful and on-point comments, critiques, and suggestions, which greatly help improve the content of our study. In the following point-by-point responses to the reviewers, the comments by the reviewers are indicated as bold and changes to the manuscript have been made in red font in the manuscript.

Funamizu et al. present an interesting single-centre retrospective analysis of 201 patients with ESRD on haemodialysis and their associated outcomes with and without statin therapy following PCI. The patient information was obtained from a PCI registry database and includes a median 2.8 year follow-up. The manuscript is well prepared and addresses an important clinical question.

We thank the Reviewer for her/his constructive feedbacks and we revised our manuscript accordingly.

Abstract

Line 17:PCI

Response: We accordingly inserted “PCI” following “who underwent”.

Line 18 - I understand what you did from reading the methodology but you say 2 groups and then list 4

Response: We apologize for the confusion. In this study, we divided participants into 2 groups by different 2 factors (with/without statins and low/high LDL level). Accordingly, we revised the sentence in the abstract as follows.

“Participants were divided into 2 groups by following 2 factors, such as 1) with or without statin, and 2) with or without high LDL-C (> and ≤ LDL-C = 93 mg/dL, median) at the PCI procedure.”

Line 19 – at the time of PCI

Response: we revised the abstract accordingly.

Line 20 – not that clear

Response: Again, we apologize the confusion. To be the definitions of endpoints in this study clear, we revised the abstract as follows.

The primary endpoint was defined as CV death, and the secondary endpoints included all-cause and non-CV death, and 3 point major cardiovascular adverse events (3P-MACE) which is the composite of CV death, non-fatal myocardial infarction and stroke (line 20-22).

Line 25 – grammar

Response: divided in accordance with→”divided by” (line 25)

Line 38 – which is associated with

Response: we accordingly revised the introduction by putting “is associated with” instead of “associated” (line 39).

Line 54 – check grammar

Response: We thank the reviewer for his/her suggestion. We revised the sentence as follows (line 53-54).

Nevertheless, since these studies did not include many patients having both ESRD on dialysis and CAD, the merit of statins is still open to question in such particular population.

Supplementary Figure S1 – I would be interested to see the loss to follow up (your median follow up was 2.8 year to and range 0-15.2 can be broken down) numbers on the CONSORT diagram. For consistency, first box of CONSORT, should be (n=4542)

Response: In this study and registry database, follow-up was done described in the methods section, “In this prospective PCI registry database, the follow-up of patients was based on chart review, as far as they were followed at Juntendo University Hospital. A prognosis survey questionnaire was mailed out every 5 years if they were transferred to other institutions. When there was no response to the questionnaire, further follow-up information was obtained by telephone. In cases in which no response could be obtained by either mail or phone, follow-up was terminated at the latest time point at which their survival at our institution was confirmed, such as the last visit date to an outpatient clinic or the last day of any hospitalization”. In this method, follow-up duration is the same as that of the length of hospitalization, when the follow-up was terminated right after the discharge from the hospital. Therefore, theoretically, there is no patient who was lost to follow-up in this registry database. 

Methods Page 2, Line 83 - Follow up – was compliance to statin therapy assessed? I would think that this is of high importance provide the paper is investigating outcomes with and without statin therapy. Furthermore, were the duration of anti-platelet therapy/warfarin following PCI from the two groups collected? 

Response: We thank the Reviewer for the insightful comments/questions regarding important points. Although we unfortunately have data of neither compliance nor duration of medications, we agree with the Reviewer that these data are important.  

Results Page 3, Line 132 – were data collected on the distributions on the types of statins used in the statin group? there has been landmark trials for specific statins so does your population differ from those previously published

Response: We have collected data regarding types of statins as attached. Nevertheless, as the sample size is limited with statins (n=73) and types are various (6 types), a subanalysis of each type of statins might be underpowered.

Reviewer 2 Report

Patients with ESRD undertaking maintenance hemodialysis is the high risk group for restenosis or CV mortality after PCI. Therapeutic modalities for the prevention of MACE are disappointing, partially can be attributed to the diffuse vascular calcification. In this essence, an effective treatment is warranted.

As the authors mentioned in discussion, there are several inherited drawbacks in this manuscript. Firstly, this is a retrospective analysis based on a single center registry database of a relatively small sample size (N = 201). Moreover, the duration, dose, potency, and adverse effects of statin administered were not specified.  

Regarding the definition of endpoints:

  1. Primary endpoint: CV death, defined as a composite of the following types of death; sudden death in which “non-cardiac death could not be excluded”, and death due to MI, HF, cardiogenic shock, and cerebral vascular event, or aortic disease. If it is the case, then some noncardiac death may be included into CV death. Could the author clarify this sentence?
  2. The Secondary endpoints: included all cause death, “mortality other than CV-related (non-CV death)” . Please give the definition of “non-CV death”. If it is defined by exclusion of the primary endpoint, then some non-cardiac death may be excluded.

Supplement table S3: Univariate-unadjusted Cox proportional hazard analysis for predicting CV mortality, all-cause mortality, and 3P-MACE. The TG level significantly correlated with HR of CV death, all-cause mortality, and 3P-MACE. Could the author give some explain on this phenomena. Does the TG level correlate with blood sugar level or other metabolic derangement? 

Author Response

The authors sincerely thank the Editors and Reviewers for providing us the opportunity to revise our study, as well as for a number of insightful, helpful, and on-point comments, critiques, and suggestions, which greatly help improve the content of our study. In the following point-by-point responses to the reviewers, the comments by the reviewers are indicated as bold and changes to the manuscript have been made in red font in the manuscript.

Reviewer 2

Patients with ESRD undertaking maintenance hemodialysis is the high risk group for restenosis or CV mortality after PCI. Therapeutic modalities for the prevention of MACE are disappointing, partially can be attributed to the diffuse vascular calcification. In this essence, an effective treatment is warranted.

As the authors mentioned in discussion, there are several inherited drawbacks in this manuscript. Firstly, this is a retrospective analysis based on a single center registry database of a relatively small sample size (N = 201). Moreover, the duration, dose, potency, and adverse effects of statin administered were not specified.  

Regarding the definition of endpoints:

  1. Primary endpoint: CV death, defined as a composite of the following types of death; sudden death in which “non-cardiac death could not be excluded”, and death due to MI, HF, cardiogenic shock, and cerebral vascular event, or aortic disease. If it is the case, then some noncardiac death may be included into CV death. Could the author clarify this sentence?
  2. The Secondary endpoints: included all cause death, “mortality other than CV-related (non-CV death)”. Please give the definition of “non-CV death”. If it is defined by exclusion of the primary endpoint, then some non-cardiac death may be excluded.

Response: We thank the reviewer for this on-point question. In the present study, we used the endpoint definitions in accordance with the consensus report published by the United States Food and Drug Administration (US-FDA) (Circulation. 2018;137:961–972), which described that a common analytic approach is to assume that all undetermined deaths are CV deaths. However, as the Reviewer is concerned, the definitions of CV or non-CV death might include deaths of opposite definitions To avoid confusion, we put the FDA consensus report as a reference.

Supplement table S3: Univariate-unadjusted Cox proportional hazard analysis for predicting CV mortality, all-cause mortality, and 3P-MACE. The TG level significantly correlated with HR of CV death, all-cause mortality, and 3P-MACE. Could the author give some explain on this phenomena. Does the TG level correlate with blood sugar level or other metabolic derangement?

Response: This is another very insightful comments and critiques. We actually have been interested in the association between reduced triglycerides and poor outcomes in patients following PCI, which challenges the long-standing dogma of elevated triglycerides, an established cardiovascular risk. As consequence, we found that very low level of triglycerides might be caused by poor nutritional status, which leads to poor cardiovascular outcomes, then we invented a novel nutritional index including blood triglycerides, body weight, and total cholesterol level which well reflects nutritional status and is a good prognostic indicator in PCI patients (Int J Cardiol. 2018;262:92-98.), in patients with cardiovascular assist devices (Nutrients. 2019 11(6):1420.), and in patients with heart failure (Nutrients 2020, 12(11), 3311).

Reviewer 3 Report

The aim of the present study was to explore whether statin therapy is associated with a reduced risk of adverse CV outcomes in hemodialysis patients following PCI.

The study is overall interesting.

English revision is necessary to better understand the manuscript, for example, "this study included consecutive patients on chronic hemodialysis who underwent for the first time between 2000 and 2016 (n=201)" it is not clear what the authors intended to express here.

"Participants were divided into 2 groups 1) with or without statin, and 2) with or without high LDL-C (> and ≤ LDL-C = 93 mg/dL, median) at the PCI procedure". Why did the authors use this value of LDL-C = 93 mg/dL?

The authors should define each abbreviation at the first appearance in the text, even in the abstract. For example "3P-MACE"

"Table 1: Baseline clinical characteristics of the study population" should be divided into more tables.

In the results, the authors present some data about blood tests but it is not mentioned when the blood samples were taken

The results section and discussion section are well written.

Author Response

The authors sincerely thank the Editors and Reviewers for providing us the opportunity to revise our study, as well as for a number of insightful, helpful, and on-point comments, critiques, and suggestions, which greatly help improve the content of our study. In the following point-by-point responses to the reviewers, the comments by the reviewers are indicated as bold and changes to the manuscript have been made in red font in the manuscript.

The aim of the present study was to explore whether statin therapy is associated with a reduced risk of adverse CV outcomes in hemodialysis patients following PCI.

The study is overall interesting.

Response: We thank the Reviewer for his/her positive and constructive comments on our study.

English revision is necessary to better understand the manuscript, for example, "this study included consecutive patients on chronic hemodialysis who underwent for the first time between 2000 and 2016 (n=201)" it is not clear what the authors intended to express here.

Response: We apologize for the confusion. Accordingly, we inserted “PCI” following “who underwent” (line 17).

"Participants were divided into 2 groups 1) with or without statin, and 2) with or without high LDL-C (> and ≤ LDL-C = 93 mg/dL, median) at the PCI procedure". Why did the authors use this value of LDL-C = 93 mg/dL?

Response: For dividing participants equally, we used the median value (93mg/dl) of LDL-C value.  

The authors should define each abbreviation at the first appearance in the text, even in the abstract. For example "3P-MACE"

Response: We spelled out 3P-MACE in the abstract as “3 point major cardiovascular adverse events (3P-MACE) which is the composite of CV death, non-fatal myocardial infarction and stroke.” (line 21-22).

"Table 1: Baseline clinical characteristics of the study population" should be divided into more tables.

Response: We agree with the Reviewer that the volume of information in Table 1 might be too high, but the number of Tables is limited. Therefore, subtitles in the table, such as laboratory findings and medications, are inserted to make it easy to see.

In the results, the authors present some data about blood tests but it is not mentioned when the blood samples were taken

Response: We inserted a sentence regarding blood sampling at the bottom of the Participants section “Blood samples were collected in the early morning after overnight fasting” (line 83-4).

The results section and discussion section are well written.

Again, we appreciate the Reviewer’s comments which are greatly helpful to improve the contents of our study.

Reviewer 4 Report

There is no novelty of the research. It was previously investigated. 

Author Response

The authors sincerely thank the Editors and Reviewers for providing us the opportunity to revise our study, as well as for a number of insightful, helpful, and on-point comments, critiques, and suggestions, which greatly help improve the content of our study. In the following point-by-point responses to the reviewers, the comments by the reviewers are indicated as bold.

Response: We are very sorry that we could not make a clear presentation regarding the significant novelty in our study for this Reviewer. Nevertheless, we believe that it clarified the prognostic merit of statins in patients who both have end-stage renal disease on maintenance hemodialysis and established coronary artery disease following PCI. More interestingly, it indicated for the first time that the beneficial effect of statins might be mediated by a property other than LDL-C lowering, such as the anti-inflammatory effect of statins. Anyways, we authors would like to appreciate the time, efforts, and comments regarding our study by the Reviewer.